# Exploring Real World Outcomes with Nivolumab Plus Ipilimumab in Patients with Metastatic Extra-Pulmonary Neuroendocrine Carcinoma (EP-NEC)

**DOI:** 10.3390/cancers14112695

**Published:** 2022-05-30

**Authors:** Amr Mohamed, Namrata Vijayvergia, Matthew Kurian, Lisa Liu, Pingfu Fu, Satya Das

**Affiliations:** 1Department of Medicine, Division of Hematology and Medical Oncology, University Hospitals, Seidman Cancer Center, Case Western Reserve University, Cleveland, OH 44106, USA; matthew.kurian@uhhospitals.org (M.K.); pxf16@case.edu (P.F.); 2Department of Hematology and Oncology, Fox Chase Cancer Center, Philadelphia, PA 19111, USA; namrata.vijayvergia@fccc.edu (N.V.); lisa.liu@fccc.edu (L.L.); 3Department of Medicine, Division of Hematology and Medical Oncology, Vanderbilt University Medical Center, Nashville, TN 37232, USA; satya.das@vumc.org

**Keywords:** neuroendocrine carcinoma, chemotherapy, checkpoint inhibitor therapy, nivolumab and ipilimumab

## Abstract

**Simple Summary:**

Extrapulmonary neuroendocrine carcinomas (EP-NEC) are a group of tumors which are often metastatic and characterized by poor outcomes. Platinum-etoposide chemotherapy is the current front-line therapy for metastatic EP-NEC, and has been adapted from small cell lung cancer. There are limited treatment options for patients with platinum-resistant EP-NEC, with no current established second-line standard of care. Recently, there has been mixed evidence for the role of immunotherapy in EP-NEC, with limited existing prospective data. In this multicenter retrospective analysis, we compared outcomes between patients with refractory EP-NEC who received single, dual immune check point inhibitors (ICPIs) and cytotoxic chemotherapy in the second-line setting. This real world experience suggests that utilizing ipilimumab and nivolumab in patients with second-line pretreated EP-NEC may be more effective than other existing treatment options.

**Abstract:**

*Background*: Dual utilization of the immune checkpoint inhibitors (ICPIs) nivolumab plus ipilimumab has demonstrated clinical promise in the treatment of patients with refractory high-grade neuroendocrine neo-plasms (NENs) in phase II clinical trials (DART SWOG 1609 and CA209), while single agent ICPIs have largely been ineffective for these types of tumors. While both trials demonstrated promising results in high grade NENs, there was no adequate description of the association between tumor differentiation (high-grade well-differentiated neuroendocrine tumor vs poorly-differentiated extra-pulmonary neuroendocrine carcinoma (EP-NEC) and ICPI outcomes in the DART SWOG 1609 trial. Our study reports on the effectiveness and toxicity profile of dual ICPIs in a real world second-line EP-NEC patient population. *Methods*: Data on metastatic EP-NEC patients, treated with either ICPIs (single and dual ICPIs) or chemo-therapy in the second-line setting, were retrieved from databases of three comprehensive cancer centers. Associations between treatment characteristics and outcomes, including progression-free survival (PFS) and overall survival (OS), were evaluated. *Results*: From 2007 to 2020, we identified 70 patients with metastatic EP-NEC (predominantly of gastro-enteropancreatic origin), of whom 42 patients (23 males, 19 females, median age 62 years old) were eligible for the final analysis. All patients were refractory to platinum etoposide doublet chemotherapy in the first-line setting. The median PFS for patients who received dual ICPIs (11 patients), single agent ICPI (8 patients), and cytotoxic chemotherapy (23 patients) was 258 days, 56.5 days, and 47 days, respectively (*p* = 0.0001). Median overall survival (OS) for those groups was not reached (NR), 18.7 months, and 10.5 months, respectively (*p* = 0.004). There were no significant differences in treatment outcomes in patients according to tumor mismatch repair (MMR) or tumor mutational burden (TMB) status. Grade 3–4 adverse events (AEs) were reported in 11.1% of the patients who received dual ICPIs; however, none of these AEs led to permanent treatment discontinuation. *Conclusions*: In the second-line setting, patients with EP-NECs treated with dual ICPIs (nivolumab plus ipilimumab) experienced improved PFS and OS compared to patients treated with single agent ICPI or cytotoxic chemotherapy. These results echo some of the current evidence for ICPIs in grade 3 NENs and need to be validated in future prospective studies.

## 1. Introduction

Poorly-differentiated extra-pulmonary neuroendocrine carcinomas (EP-NECs) are a group of neoplasms with highly aggressive behavior, defined by ki-67 proliferative index more than 20% and limited therapeutic options [1,2,3]. Treatment for EP-NECs has been adapted from small cell lung cancer (SCLC), with platinum etoposide doublet chemotherapy utilized as the standard first-line regimen [4,5]. In the later-line setting, very few treatments have been prospectively validated in patients with EP-NECs, with most treatment recommendations stemming from small retrospective studies. There are very few data on second-line therapy for patients progressed on platinum etoposide, with no established standard of care. Previous retrospective studies have demonstrated modest benefit from oxaliplatin- or irinotecan-based cytotoxic chemotherapy. Other data suggested topotecan as a second-line treatment in GEP NEC patients, with poor response rate and overall survival.

Immune checkpoint inhibitors (ICPIs) have demonstrated antitumor activity in many solid tumors; however, their role in the treatment of NENs has only recently been investigated. Several clinical trials of (ICPIs) have been completed in patients with EP-NECs. The results from these trials have demonstrated a limited role for single agent ICPIs in patients with EP-NECs with low objective response rate (ORR) (3–5%), short progression-free survival (PFS) (less than four months), and limited overall survival (OS) [6,7,8,9]. The limited activity of single agent immune checkpoint blockade in metastatic EP-NECs particularly of gastrointestinal origin has been attributed to low tumor mutational burden, lymphocyte infiltration and lack of programmed death-ligand-1 (PD-L1) expression. Given the suboptimal anti-tumor activity of single agent ICPIs in EP-NECs, approaches with dual ICPIs that target programmed cell death 1 (PD-1) and cytotoxic T-lymphocyte-associated protein 4 (CTLA-4) have been evaluated. Treatment with dual ICPIs has led to dramatic survival improvements for patients with many solid tumors; however, the data for their utilization in EP-NECs is still limited. Two recent phase II trials (DART and CA209-538) evaluated the anti-tumor activity of dual anti-CTLA-4 and anti-PD-1 blockade in high-grade neuroendocrine neoplasms [10,11]. Despite promising results, there were significant limitations to each trial. In the DART trial, there was no adequate description of tumor differentiation, while the CA209-538 trial enrolled a very small number of poorly-differentiated NECs (only two patients possessed GEP-NECs). Additionally, single institution experiences and real world data have shown inconclusive results regarding the benefit of dual ICPIs for patients with EP-NECs in the second-line setting [12,13,14]. A phase II study in this patient population [NCT04079712] has not yet reported any results. Our study evaluated real world data from three comprehensive cancer centers to better understand the effectiveness and toxicity profile of dual ICPIs in patients with refractory EP-NECs.

## 2. Materials and Methods

A retrospective study of patients with metastatic EP-NECs who were treated with either ICPIs (single and dual) or chemotherapy in the second-line setting between January 2007 and June 2020 was initiated. The records of these patients were retrieved from the databases of three comprehensive cancer centers (Seidman Cancer Center, Cleveland, OH, USA; Vanderbilt-Ingram Cancer Center, Nashville, TN, USA; and Fox Chase Cancer Center, Philadelphia, PA, USA). Institutional review board approval for sharing de-identified data was obtained from each center with a waiver of consent due to the study’s retrospective nature. We included patients with a histological diagnosis of EP-NEC (mostly of gastroenteropancreatic origin) with prior progression on one prior line of treatment, consisting of platinum etoposide doublet chemotherapy, who received some form of systemic therapy (immunotherapy, chemotherapy, or targeted agents) in the second-line setting. We collected demographic, clinical, and pathologic data including age, sex, race, primary site of the tumors, histological differentiation, and grade identified with ki-67% and/or mitotic index, prior and current oncologic treatments, date of diagnosis, date progression on each line of therapy, date of last follow up, and date of death. We also collected next-generation sequencing results when available. We excluded patients with well-differentiated neuroendocrine tumors, lung NECs, and Merkel cell carcinomas. We also excluded patients who did not receive first-line platinum etoposide doublet chemotherapy, and those who received an undefined second-line therapy. Associations between treatment characteristics and outcomes, including median PFS, OS, and treatment-related toxicity were evaluated using retrospective chart review. The primary endpoint of this study was PFS, which was measured from the date of treatment start to the date of disease progression or the date of death, whichever occurred first. Secondary endpoints included OS, which was measured from the date of onset of treatment to the date of death and rates of grades 3–4 treatment related toxicities. In-addition, medical records were reviewed to identify patients underwent genomic sequencing, and results were extracted for analysis. The disease progression and survival data were estimated by Kaplan-Meier method and categorical variables were analyzed using logistic regression. A significance level of 0.05 was used for final analyses.

## 3. Results

Seventy patients with metastatic poorly-differentiated EP-NEC were identified from the database of three cancer centers during the period of 2007–2020. Twenty-eight patients did not meet criteria for inclusion and were excluded (reasons included a lack of receipt of platinum etoposide doublet chemotherapy in the first-line setting, a lack of available data for second-line therapy received, or ineligibility for second-line therapy due to poor performance status). Forty-two patients (23 males, 19 females; median age 62 years) met the inclusion criteria for evaluation and were eligible for the final analysis. All patients possessed metastatic poorly-differentiated EP-NEC. The primary tumor was identified as originating from gastroenteropancreatic origin for most patients (*n* = 35), while seven patients possessed a tumor of unknown primary. The most common primary sites were small bowel (*n* = 13), pancreatic (*n* = 11), colorectal (*n* = 8), unknown primary (*n* = 7), and other GI sites such as gall bladder, gastroesophageal (*n* = 3). Median ki-67 was 70% (ki-67 index range from 30–98%). Patient characteristics are described in detail in Table 1.

In the second-line setting, 11 patients received dual ICPIs (nivolumab plus ipilimumab), eight patients received single agent ICPI (nivolumab or pembrolizumab), and 23 patients were treated with cytotoxic chemotherapy (4% taxanes and topoisomerase 1 inhibitors, 39% irinotecan-based, and 8.6% oxaliplatin based regimens) or targeted therapies (48.4% including everolimus, olaparib, and sunitinib). Dosing and schedule for dual ICPI therapy varied between sites and subjects based on the DART and CA209 trial schedules [10,11].

The median PFS for patients who received dual ICPIs compared with median PFS in patients who received single ICPIs and chemotherapy/targeted therapy was 258 days, 56.5 days and 47 days respectively (*p* = 0.0001) (Figure 1). Median OS for patients who received dual ICPIs compared with median OS in patients who received single ICPIs and chemotherapy/targeted therapy was not reached (NR), 18.7 months, and 10.5 months, respectively (*p* = 0.004) (Figure 2). There was insufficient genomic information to demonstrate significant differences in treatment outcomes according to tumor PD-L1 expression, tumor mismatch repair (MMR) status, or tumor mutational burden (TMB) status. Grade 3–4 adverse events (AEs) were reported in 11.1% of the patients who received dual ICPIs; with fatigue and transaminitis being the most common toxicities. None of these AEs led to treatment discontinuation. The most common grade 3–4 toxicities for those who received chemotherapy or targeted therapy included nausea, vomiting, fatigue, and myelosuppression. Toxicity from patients receiving ICPIs is described in Table 2.

## 4. Discussion

Current data regarding second-line therapy for patients with refractory EP-NECs are limited, and are based on scant retrospective data and case series [15,16,17,18,19]. Most of this data are derived from data on patients treated with standard chemotherapy regimens, including topotecan, FOLFOX, or FOLFIRI. In aggregate, among patients with EP-NECs treated with topotecan, median PFS was roughly 2 months, with median OS times of 3–4 months. Among patients with EP-NECs treated with FOLFOX or FOLFIRI, median PFS and median OS approximated 4 and 9–18 months, respectively. Most patients who progress on platinum etoposide doublet chemotherapy have few therapeutic options, which are largely associated with suboptimal benefit. Similarly, the role of the single agent ICPIs is also not established in patients with EP-NECs. Multiple small prospective studies with single agent PD-1 inhibitors have consistently reported low ORR (3–5%) and PFS (less than 4 months) [6,7,8,9]. Given the suboptimal efficacy of chemotherapy or targeted therapy and single agent ICPIs in refractory EP-NEC, two phase II studies evaluated the benefit of dual anti-CTLA-4 and anti-PD-1 blockades, with promising results but significant limitations, leaving room for debate and speculation about the actual role for dual ICPIs in this disease [10,11]. Subsequently, results from the DUNE trial were also reported. In this multi-cohort phase II study, 123 patients with advanced NENs of gastroenteropancreatic or lung origin were treated with the anti-PD-L1 inhibitor durvalumab plus the anti-CTLA 4 inhibitor tremelimumab. The study enrolled patients with both well- and poorly-differentiated NENs [20]. In the poorly-differentiated NEN cohort (90% with poorly-differentiated tumors), 9-month OS was 36.1%; this was the only cohort in which the prespecified threshold for futility was not exceeded. The ORR in this cohort was 7.2%. Durvalumab plus tremelimumab demonstrated modest activity in heavily pretreated patients, with higher grade NENs; however, again, a mixed population of patients with well-differentiated grade 3 neuroendocrine tumors (NETs) and PD-NECs was included.

Our study suggests that therapy with the dual ICPIs nivolumab plus ipilimumab treatment demonstrates reasonable activity in the second-line setting in patients with poorly-differentiated EP-NECs, with improved PFS and OS compared to both single agent ICPIs and chemotherapy/targeted therapy [10,11]. Our study results represent a real world experience and highlight the benefit of dual ICPIs in NECs where prospective randomized data are limited, pending, and conflicting. Toxicities reported in our data set were also similar to previously reported data (approximately 10%) [10,11,21,22].

Insufficient genomic data precluded the assessment relationship between tumor TP53, Rb1, PDL-1, MMR, and TMB status and treatment outcomes. Although PD-L1 expression has been widely used in many solid tumors to select the patients who may benefit disproportionately from ICPI therapy, its predictive value in EP-NEC is still unclear [6,7,9]. Other predictive biomarkers for ICPI responsiveness, such as MMR and TMB status, have demonstrated predictive capacity for patients across multiple malignancies; however, their role in predicting benefit for EP-NEC patients remains to be determined [23,24,25].

Our study has some primary limitations including its retrospective nature and small sample size. Other limitations include the lack of a second-line systemic standard of care for patients with EP-NECs to contextualize our findings; in addition, there was a lack of uniformity with regards to the chemotherapy and targeted therapy received by patients who did not receive ICPIs.

Despite these limitations, this is one of the largest retrospective studies to date investigating the role of dual ICPIs in patients with poorly-differentiated EP-NEC, and adds to the body of literature studying these agents in this setting. The anti-tumor activity demonstrated in patients with EP-NECs suggests that this regimen possesses promise for this patient group. Ongoing and future studies further evaluating addition of ICPIs to the treatment paradigm of EP-NECs are of paramount importance. The successful addition of immunotherapy to first-line platinum plus etoposide in metastatic SCLC has prompted this approach to be explored in extra-pulmonary small cell carcinoma [NCT 05058651].

In previous clinical trials of ICPIs in high grade NENs, there was limited information regarding predictive biomarkers to guide patient selection for immunotherapy. A critically important factor, to even have the opportunity to define some of these biomarkers in NECs, is separating well-differentiated NETs from PD NECs in studies. Only over the last decade have patients with NECs been more consistently studied individually [26]. One of the biomarkers thought to be predictive of response to ICPIs in other solid tumors includes the degree of tumor infiltrating lymphocytes (TILs) [27,28]. Currently published data suggest that the majority of well-differentiated NETs are “immunologically cold”, with low tumor mutational burden and rare TILs compared to poorly-differentiated NECs, which have a higher degree of TILs [29,30]. Additionally, the genetic profile of poorly-differentiated NEC differs vastly from the genetic profile of well-differentiated NETs. The former is characterized by TP53 and Rb1 mutations while the latter (particularly those of pancreatic origin) is characterized by DAXX and ATRX mutations [31,32]. Given these differences, it is highly plausible that there are differing predictive biomarkers of response to ICPIs in each of these patient populations. We believe it is important, in future prospective studies of ICPs in patients with EP-NECs, to collect tissue and blood based biomarkers in order to gather more adequate date of potential biomarkers for immunotherapy response.

## 5. Conclusions

In summary, we observed improved PFS and OS outcomes in patients with EP-NECs refractory to platinum etoposide treated with the dual ICPIs nivolumab plus ipilimumab compared to single ICPIs or cytotoxic chemotherapy in the second-line setting. In the absence of standard of care options or available clinical trials, dual ICPIs can be considered off-label as a potential second-line treatment option for this group of patients. Definitive evidence from prospective trials, however, are needed to verify the benefit of dual ICPIs in patients with EP-NECs.

## Figures and Tables

**Figure 1 cancers-14-02695-f001:**
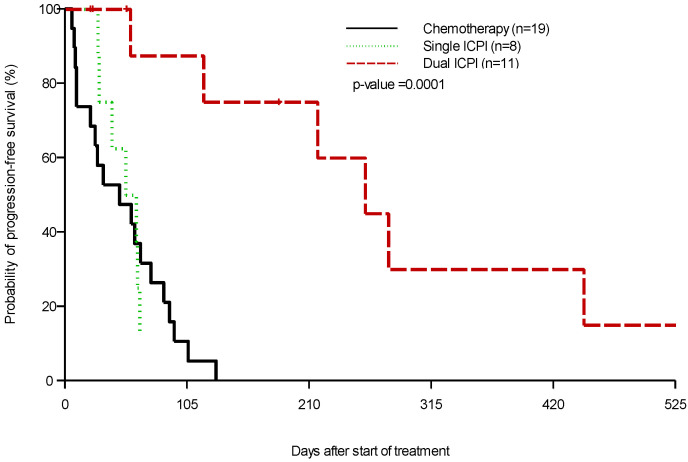
Kaplan–Meier estimation of progression-free survival by treatment group. Available median PFS for patients who received dual ICPIs compared with single ICPIs and chemotherapy/targeted therapy (258 days, 56.5 days, and 47 days, respectively (*p* = 0.0001).

**Figure 2 cancers-14-02695-f002:**
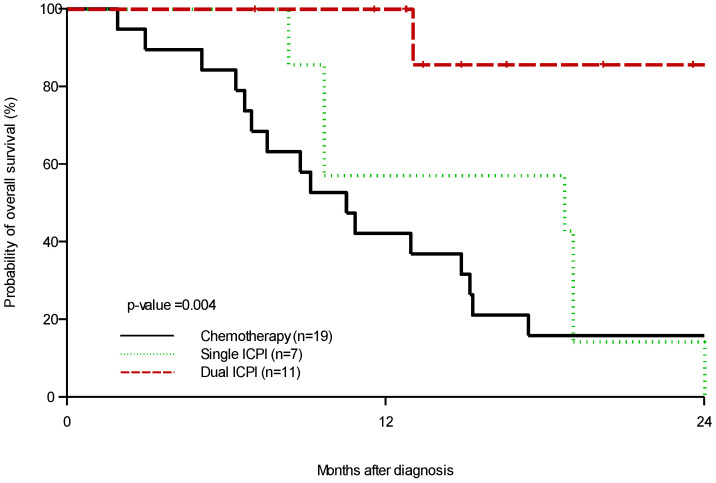
Kaplan–Meier estimation of overall survival by treatment group. Available median OS for patients who received dual ICPIs compared with single ICPIs and chemotherapy/targeted therapy (not reached (NR), 18.7 months, and 10.5 months, respectively (*p* = 0.004).

**Table 1 cancers-14-02695-t001:** Patient demographics and treatment characteristics.

Characteristics	ICPIs *n* = 19	Chemotherapy *n* = 23	*p*-Value
Age (Median)	63	59	
Male %	57%	48%	
Primary Site (*n*)			
Pancreatic	7	4	
SB	3	10	
Colorectal	2	6	
Other GI	1	2	
Unkown	6	1	
Median ki-67%	71%	79%	
Most common sites of metastases	Liver, Lymph nodes, pulmonary, and osseous metastases	Liver, Lymph nodes, and osseous metastases	
PFS (Days)	Dual ICPIs: 258 Single ICPI: 56.5	47	0.0001
OS (Months)	Dual ICPI: NR Single ICPI: 18.7	10.5	0.004

**Table 2 cancers-14-02695-t002:** Adverse Events from Patients Receiving Therapy with Immune Checkpoint Inhibitors.

Adverse Event	Grade ½ (*n* = Number)	Grade ¾ (*n* = Number)
Fatigue	11	1
Transaminitis	1	1
Anorexia	3	2
Nausea	3	
Abdominal Pain	2	
Anorexia	3	1
Back Pain	1	
Diarrhea	2	
Pulmonary Embolism		1

## Data Availability

The data presented in this study are available on request from the corresponding author.

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
