# Peer review of "Exploring Real World Outcomes with Nivolumab Plus Ipilimumab in Patients with Metastatic Extra-Pulmonary Neuroendocrine Carcinoma (EP-NEC)"

_cancers, 2022, doi:10.3390/cancers14112695_

Round 1
Reviewer 1 Report
I agree with the authors that the role of dual check point inhibitor therapy needs further evaluation. The results of this well constructed retrospective analysis further supports that.
On review of the content:
abstract: covers the background, methods, and results succinctly. conclusion adequate
background: includes the most up to date analysis available regarding IO therapy in NET/NEC.
methods: clear. IRB approved multi-center retrospective analysis. statistical considerations were adequate.
results: clear.
conclusions: results suggest a benefit. Agree that these results support further investigation but appreciate that the authors tempered recommendations that this should be considered SOC. Hopeful the current prospective cooperative group study will successfully answer this question.
- Although there was no standard 2nd line cytotoxic regimen, was there an analysis performed comparing the various options utilized? I know not the focus of this analysis but may be worth mentioning is a notable % was treated with one regimen.
- What percentage of patients had molecular data available? if there was a subset, was there an analysis regarding a correlation between the profiles available and clinical outcomes? You mention TILs in your conclusion, was this information available on any of your patients?
- minor grammatical finding. line 116, unkown should be corrected to unknown
Author Response
- Although there was no standard 2nd line cytotoxic regimen, was there an analysis performed comparing the various options utilized? I know not the focus of this analysis, but may be worth mentioning is a notable % was treated with one regimen.
- The chemotherapy group consisted of 23 patients. The types of chemotherapy and targeted therapies were highly variable and consisted of 48.4% targeted therapies such as: Everolimus, Olaparib and Sunitinib, 39% Irinotecan based chemotherapy (FOLFIRI), 8.6% Oxaliplatin based (FOLFOX), and 4% received Taxanes or Topotecan.
- Due to lack of consistency and high variability, we did not compare between different types of chemotherapies and targeted therapies used in the second line setting.
- What percentage of patients had molecular data available? If there was a subset, was there an analysis regarding a correlation between the profiles available and clinical outcomes? You mention TILs in your conclusion, was this information available on any of your patients?
- 17/42 (40%) of the patients had some genomic data available. Only 3 patients had information available about TMB (2 patients had high TMB and only one patient had MSI-H) with not enough information about TLS. Therefore, genomic data was inconsistent and insufficient to drive significant correlation with clinical outcomes.
- Minor grammatical finding. Line 116, unknown should be corrected to unknown.
- Thank you for your comment. This minor grammatical finding has been corrected.

Reviewer 2 Report
The work by Mohamed and co-authors is very important and offers new information about immunotherapy in patients with extra-pulmonary NEC. Studies handling this topic are ongoing. But mostly the patient numbers are low. However, these results demonstrate evidence that this therapy could be promising.
Introduction:
The general part about EP-NECs could be more explained to the readers e.g. prognosis, alternative treatment options
Results:
How many patients received pembrolizumab and targeted therapies in the chemotherapy group. This should be clarified.
Please describe the TMB, MMR and PD-L1 expression better as mentioned in line 131-132. Any pts. MSI high or TMB more than 5/10?
The demographics and treatment characteristics table can be improved. Please include ECOG, median Ki-67, sites of metastasis, prior resection…
Why not include the 28 pts. which did not meet the criteria, as a separat population. Could be interesting which kind of therapy EP-NEC received when not platinum-based.
The toxicities need explained in a separate table more accurately then performed.
The quality and description of the figures is nice.
Discussion:
Good structured and easy to read.
Author Response
Reviewer 2:
- Introduction: The general part about EP-NECs could be more explained to the readers e.g. prognosis, alternative treatment options.
- Thank you for your comment. A paragraph discussing NEC and available therapeutic options in refractory setting has been added in the introduction.
- Results: How many patients received pembrolizumab and targeted therapies in the chemotherapy group? This should be clarified.
- None of the patients in the chemotherapy group had received pembrolizumab.
- Approximately 48.4% received targeted therapies including the following: Everolimus, Olaparib, and Sunitinib. All of which has been added to the results section.
- Please describe the TMB, MMR and PD-L1 expression better as mentioned in line 131-132. Any pts. MSI high or TMB more than 5/10?
- Only 3 patients had information available about TMB (2 patients had high TMB and only one patient had MSI-H) and not enough information to drive significant correlation with clinical outcomes.
- The demographics and treatment characteristics table can be improved. Please include ECOG, median Ki-67, sites of metastasis, prior resection.
- The table has been improved with the available information including median Ki67 and site of metastasis.
- Why not include the 28 pts? Which did not meet the criteria, as a separate population? Could be interesting which kind of therapy EP-NEC received when not platinum-based.
- I think, while an interesting point raised by the reviewer, patients not treated with platinum-based chemotherapy traditionally have very poor treatment outcomes. Their post-progression status would be different and introduce a significant degree of heterogeneity to the analysis.
- The toxicities need explained in a separate table more accurately then performed.
- We have now included a Table 2 which specifically outlines the adverse events experienced by patients receiving immune checkpoint inhibitors.

Round 2
Reviewer 2 Report
The points were conclusively discussed. No further comments from my site.